# Agricultural Application Prospect of Fully Polarimetric and Quantification S-Band SAR Subsystem in Chinese High-Resolution Aerial Remote Sensing System

**DOI:** 10.3390/s24010236

**Published:** 2023-12-31

**Authors:** Yabo Liu, Luhao Wang, Shuang Zhu, Xiaojie Zhou, Jia Liu, Binghong Xie

**Affiliations:** 1Aerospace Information Research Institute, Chinese Academy of Sciences, Beijing 100094, China; wangluhao22@mails.ucas.ac.cn (L.W.); zhushuang21@mails.ucas.ac.cn (S.Z.); xiebh@aircas.ac.cn (B.X.); 2School of Electronic, Electrical and Communication Engineering, University of Chinese Academy of Sciences, Beijing 100049, China; 3School of Electronic Information Engineering, University of Beihang, Beijing 100083, China; zhouxiaojie0730@buaa.edu.cn; 4Institute of Agricultural Resources and Regional Planning, Chinese Academy of Agricultural Sciences, Beijing 100089, China; liujia06@cass.cn

**Keywords:** fully polarimetric SAR, calibration, quantitative, agricultural application

## Abstract

The synthetic aperture radar (SAR) is a type of active radar that can obtain polarization scattering information of ground objects, which is an important supplement to optical remote sensing. This paper designs a high-precision quantitative SAR system that combines radiation and polarization calibration processing to achieve a subtle perception of the changes in soil moisture and straw coverage. In Yushu, Jilin, we conducted the first S-band agricultural remote sensing application experiment. The backscattering coefficient was measured under different water content and straw coverage conditions, and the results showed that the backscattering coefficient increased by about 2 dB and 6 dB, respectively. We estimated that the soil water content increased by about 0.01 cm^3^/cm^3^, which was consistent with the theoretical analysis. The polarization scattering characteristics also showed significant differences under different straw coverage. The results indicated that S-band quantitative SAR had an excellent response ability to water content and straw coverage, which provided a technical basis for further radar agricultural applications in the future.

## 1. Introduction

### 1.1. Microwave Remote Sensing of Black Soil in Northeast China

Northeast China, one of the world’s four major black soil regions, covers about 170,000 square kilometers. This region produces 22.5 to 25 billion kilograms of commodity grain annually, which contributes significantly to China’s agricultural development. Soil moisture in Northeast China’s black soil is crucial for plant and crop survival, influencing plant height and crop root system development. However, after years of development and reclamation, the black soil in Northeast China faces soil erosion and ecological deterioration. The soil organic matter content has declined by nearly two-thirds compared to the pre-reclamation level, and the problems of soil compaction and salinization are severe. Consequently, quantifying soil moisture is vital for both monitoring the environment and promoting sustainable agricultural development.

Remote sensing information, with its extensive coverage, brief observation periods, and high timeliness, effectively monitors straw cover ratios, crop height, biomass, and leaf area index. Additionally, not only is it cost-effective, efficient, and precise but it also serves as a crucial technical method for examining the effects of conservation farming and straw return on soil moisture, organic matter, and physicochemical properties [1]. Optical agricultural remote sensing, an early yet relatively mature method widely used in agriculture, is limited by environmental factors, such as clouds, rain, fog, and haze, that hinder its ability to continuously observe the Earth. Microwave remote sensing can capture high-resolution images of areas of interest throughout the day. It enables continuous monitoring during critical crop growth periods and explores electromagnetic wave interactions with the environment. Furthermore, it aids in assessing soil moisture, organic matter content, and agricultural production, as well as supporting crop monitoring. Hence, microwave remote sensing has emerged as a significant approach in agricultural remote sensing.

### 1.2. Research on the Application of SAR Remote Sensing in Agriculture

SAR can uniquely identify water-related information based on its data polarization characteristics, backscattering intensity, and phase characteristics, which are sensitive to soil moisture. It holds an irreplaceable role in high spatial resolution soil moisture monitoring applications, outperforming other remote sensing methods [2,3,4,5]. The current statistical method, which relies on radar data, only analyzes the backscattering coefficient and crop residual coverage and poses significant regional limitations [6]. However, as microwave remote sensing technology advances, more information has been incorporated into crop residue coverage monitoring.

Polarization information is mainly utilized for the classification of terrestrial materials. Li et al. [7,8] applied SVM classifiers to extract the rice planting area from the fully polarimetric SAR data, using 4-scene Radarsat-2 fully polarimetric images. They demonstrated that the polarimetric decomposition features achieved better classification results than linear backscattering coefficients. Moreover, McNairn et al. [9] employed multi-phase optical and SAR images to classify crops using decision trees. They also conducted classification and monitoring of corn and soybeans with TerraSAR-X and Radarsat-2 data from pre-growth crops [10]. Xu et al. [11] discussed the relationship between the polarimetric decomposition characteristics of wheat planting areas and the change in the growth cycle using the multi-phase fully polarimetric image. They also combined the backscattering coefficient with the polarimetric decomposition feature to extract the wheat region. In addition, Jiao et al. [12] classified the 19-scene Radarsat-2 images based on objects, and the experimental results showed that Freeman–Durden decomposition and Cloude decomposition had some advantages over linear backscattering coefficients. Kenduiywo et al. [13] developed a multi-phase SAR classification method based on dynamic conditions with the airport to achieve the comprehensive utilization of spatial and multi-temporal information. Li et al. [14] also used UAVSAR data to monitor and classify the scattering characteristics of major crops in the Sacramento Valley region of California, U.S.A. Clearly, polarization information has benefits for land vegetation classification and can enhance the agricultural applications of radars.

In the study of soil water inversion, the data acquisition methods include optical remote sensing, thermal infrared remote sensing, and microwave remote sensing. Optical remote sensing, which is sensitive to surface vegetation information, uses multi-spectral data (e.g., Landsat or MODIS) to build vegetation index or drought index models that indirectly reflect soil water changes [15]. However, optical remote sensing is limited by atmospheric interference and cannot provide continuous observations in all weather conditions and throughout the day. Thermal imaging, which is sensitive to surface temperature information, can invert soil moisture using surface temperature or thermal inertia. However, it has low resolution and cannot provide detailed information due to the influence of sunshine. Active microwave remote sensing (SAR), on the other hand, can provide continuous observations in all weather conditions and throughout the day and can invert soil moisture using the quantitative relationship between SAR backscattering coefficient and soil moisture. The most commonly used semi-empirical models, such as the Oh model and the Dubois model [16], can obtain relatively high precision. SAR has a relatively complete theoretical basis and a good development in soil water estimation. Multiple SAR satellites and sensors (e.g., Sentinel-1, ALOS-2, Radarsat-2) provide rich data sources, including multi-temporal data. However, SAR requires calibration operations to obtain high-resolution data because of its working mode, even though it can penetrate vegetation to obtain soil information. Therefore, this paper presents a detailed study of the design and implementation of the SAR calibration test.

### 1.3. The Basis, Work, and Arrangement

The multi-dimensional SAR [17] developed by the Chinese high-resolution aerial system project quantifies targets in multiple dimensions, including multi-band, multi-polarized, multi-temporal, and multi-angle, to obtain more complete target characteristics for scientific research and quantitative remote sensing applications in various fields. The Aerospace Information Research Institute, CAS, developed the C-band and S-band radar subsystems. The S-band fully polarimetric SAR system passed the trial sample acceptance in December 2018. The quantitative indexes of the system are relative radiometric accuracy better than 1 dB, absolute radiometric accuracy better than 2.5 dB, polarization phase imbalance ≤ 5, amplitude imbalance ≤ 1 dB (1σ), and planimetric geometric positioning accuracy better than 5 m. 

In agricultural remote sensing applications, remote sensing data fusion is a major development direction, and multi-band SAR data is an important branch. By providing different scattering excitation, multi-band SAR data can reflect the scattering characteristics and texture information of different ground objects, enhance the ground object recognition and classification ability, and facilitate the inversion and estimation of crop species, biomass, and water parameters. S-band SAR has a moderate wavelength that balances the diversity of scattering mechanisms and the magnitude of scattering intensity and enables multi-polarized design. We conducted the first agricultural demonstration application test of fully polarimetric SAR data in Yushu, Jilin Province from 24 to 27 May 2020, based on the Chinese High-resolution Aerial System and funded by the special application demonstration project. This paper details the experiment, including the quantitative system composition, the calibration test design and implementation, the fully polarimetric SAR data processing, and the data application discussion in agriculture. To obtain robust and high-precision SAR data, we designed a polymer system with a quantitative loop and combined it with SAR data radiation correction and polarization correction technology. We then carried out accurate system calibration and obtained land polarization scattering information under different straw coverages and multi-temporal soil backscattering information.

The quantitative SAR system provides multiple observation models that can offer more valuable information for agricultural remote sensing applications. Balenzano et al. [18] inverted the absolute value of soil water with an accuracy of 0.05 cm^3^/cm^3^ by using the ratio of multi-temporal radar backscattering coefficient to eliminate the effects of surface roughness and vegetation on the radar backscattering coefficient and constructing observation equations. Fully polarimetric SAR can provide more abundant information on the polarization characteristics of different ground objects, such as different types of vegetation and even rice in different periods, which is significant for ground object recognition and classification. Fully polarimetric SAR data can also utilize the phase difference between polarized information, which single-polarized SAR and other remote sensing methods lack. This phase difference can reflect the structure and shape information of ground objects and improve the classification accuracy and reliability. Turkar et al. [19] compared the ground object classification ability of full and partially polarized SAR data of C-band and L-band and found that L-band full polarized data had the highest accuracy, and that multi-band data fusion could improve the classification accuracy by 7%. Langner et al. [20] studied surface feature classification using the SAR data of JELLS-1, ALOS, and RADARSAT-1, and showed the advantages of multi-polarized and multi-angle SAR data for surface feature mapping, especially for identifying tropical rainforest and agricultural land. The experiment also explored the demonstration applications of SAR data in observing and classifying agriculture, forestry, grassland, wetland, and water surface features, as well as straw field cover [21,22,23,24,25,26]. The S-band RCS characteristic curves of different features were established through multi-angle observation, and the relationship between the polarization scattering characteristic matrix and the observation angle was analyzed [27].

This paper is organized as follows. Section 2 introduces the S-band radar subsystem. Section 3 describes the flight program in the Yushu area, including the calibration flight and the agricultural observation test, and measures the quantitative index of the system. Section 4 obtains the RCS data and the polarization scattering characteristic matrix of the features used in this flight. Finally, Section 5 is the conclusion.

## 2. High-Resolution Aerial System S-Band Radar Subsystem

The Chinese high-resolution aerial system uses multi-dimensional SAR to quantify the target in multiple dimensions, such as multi-band, multi-polarized, multi-temporal, elevation, and multi-angle. This can provide more complete target characteristics for scientific research and quantitative remote sensing applications in various fields. The S-band radar subsystem is one of the six single-band radar subsystems in the multi-dimensional SAR system. It can provide quantitative measurement information, such as fully polarimetric, multi-temporal phase, and backscattering coefficient.

### 2.1. Technical Requirements

Table 1 shows the technical indexes achieved by the S-band radar subsystem.

### 2.2. Composition and Functions

The S-band radar subsystem has two support modes. The first mode works as a subsystem of the multi-dimensional radar system, cooperating with one or more other band radar subsystems under the centralized control of the multi-dimensional radar system. The second mode works independently as a standalone radar. 

When the single band works independently, the system composition includes an antenna unit, integrated electronics unit, a power distribution unit, a mission management unit, a data recording unit signal processing unit, motion attitude measurement equipment, and a cabinet. Figure 1 shows the system units, their connections, and their external interfaces. The antenna unit is mounted under the belly of the wheel loader, while the other equipment Is installed in the cabin of the wheel loader.

## 3. Flight Design and Preparation

The agricultural observation experiment used the S-band radar subsystem mounted on the small- to medium-sized platform of Cessna 208B from 24 to 27 May 2020. Yushu City is located in the world-famous Golden Corn Belt with fertile land, which is the national key commodity grain base and the preferred place for the deployment of monitoring points. The local area is at the stage of corn budding from May to June each year. During this period, it is of great significance to use the S-band radar remote sensing technology to monitor soil moisture, straw returning, and black soil conservation in these areas, thereby providing a basis for subsequent large-scale remote sensing applications. This application–calibration flight survey covers about 700 square kilometers. The coordinates of the surveyed area are shown in Table 2.

This quantitative scheme adopts the observation mode of air–ground collaboration. In terms of flight tests, an external calibration flight is needed before formal agricultural observation to calibrate the performance of the system since the S-band radar will be mounted on a new aircraft. To observe soil moisture and crop growth, it is necessary to conduct observation in different time phases. In the ground survey, multiple ground observation points are selected evenly in the survey area, including bare land, land covered by straw, corn fields at the germination stage, wetlands, water surfaces, and other geophysical environment conditions. 

Based on the above analysis, this experiment employs two flights on different dates to achieve the multi-temporal observation task for agricultural purposes. The first flight consists of a calibration flight outside the airport, which takes up one-third of the voyage, followed by an agricultural observation experiment in the survey area. The second flight is only for the agricultural observation experiment.

### 3.1. Route Planning

The flight task consists of two subtasks, i.e., calibration flight and survey area flight. Table 3 displays the relevant information for each subtask.

As shown in Figure 2, the white area is the area to be surveyed and the red mark is the airport where the aircraft takes off and lands. The routes from 1 to 9 cover the survey area, with an average length of 40 to 41 km each. The total length of the straight line required for a flight to cover the survey area is 410 km, and the area covered is 1600 square kilometers (including the overlap). Lastly, Route 10 is the calibration flight and X and Y are the framework routes.

A general aerial aircraft has a 5 h flight time and a 3 h effective operation time with an effective flight mileage of 750 km. Therefore, one flight can cover the whole survey area and two flights can obtain two coverages of the area at different times.

### 3.2. External Calibration Field Design and Quantitative Index Determination

#### 3.2.1. Design of External Calibration Field

External calibration flight inspection is required before carrying out agricultural application experiments. The purpose of the inspection is to examine the system function and performance index by external calibration. Specifically, the amplitude-phase consistency of polarization channels depends on both the antenna and transceiver polarization channels. The S-band radar system employs a differential antenna with a symmetrical dual-polarization feed and a mixed-mode S-parameter analysis for consistent polarization radiation. Symmetrical feed points and radiation units between polarization channels ensure full polarimetric consistency. Meanwhile, high polarization isolation ensures the purity of fully polarimetric measurement data. The system’s polarization isolation depends on the antenna, T/R polarized port, and aircraft attitude measurement accuracy. The S-band radar system stimulates a highly symmetrical radiation element with differential feeding during design for high polarization isolation.

The calibration field test equipment consists of the calibrator and other devices for installing and testing it at the external field. Table 4 shows the testing equipment for the S-band radar subsystem flight test at the calibration field.

The calibration experiment field site is selected and set at a large area with flat terrain and weak ground object backscattering. Figure 3 shows the layout diagram of the calibration field angle reflector.

In Figure 3, T1–T6 are for radar range radiometric calibration and they are uniformly distributed in the mapping bandwidth. l1 is calculated accordingly. T7–T10 are for azimuth radiometric calibration and l2 is at least 100 m. T11–T13 and D1–D3 (A/B) are for polarization calibration and l3 is at least 200 m. When deploying the calibrator, the normal line of each calibrator should overlap with the connection line between the calibrator and the aircraft in the front and side view.

To obtain a high-precision relative position relationship between the aircraft and the calibrator, the ground synchronous data acquisition scheme synchronizes the position information of both using differential measurement equipment during the flight. Table 5 summarizes the relevant radar and flight parameters.

#### 3.2.2. Index and External Calibration


1.Resolution


The selected calibration field is near the runway in the airport. The ground object of the airport is bare land, and six trihedral corner reflectors are arranged with an interval of 300 m. Figure 4 shows the SAR image of the calibration field and the profile of the calibrator.

As shown in Figure 5, the measurement resolution index of multiple calibrators shows that both the image distance and azimuth resolution are less than 1 m across the swath. Additionally, the peak side lobe ratio and the integrated side lobe ratio are above 20 dB and 15 dB, respectively.
2.Radiometric Calibration

Radiometric calibration eliminates the influence of sensor characteristics, terrain, and other factors on the SAR image radiation value and makes the image reflect the real radiation characteristics of ground objects. Radiometric calibration aims to improve the quality and comparability of SAR images and provide an accurate data basis for subsequent applications, such as classification and inversion. Relative radiometric accuracy is a measure of system stability. It refers to the maximum error of the relative value of the target backscattering coefficient or radar cross-sectional area measured at different positions in the SAR image. Usually, the maximum error value is expressed in dB and is calculated by using Equation (1):(1)$$\Delta_{rad\_r}=\sqrt{\sum_{i=1}^{N_{p}}\frac{\left({\hat{\varepsilon }}_{idB}-{\bar{\varepsilon }}_{dB}\right)}{N_{p}}}$$
where ${\hat{\varepsilon }}_{idB}$ is the point target energy of the i inspection point (expressed in dB) and ${\bar{\varepsilon }}_{dB}$ is the average value of the point target energy of $N_{p}$ inspection points and its calculation equation is given by Equation (2):(2)$${\bar{\varepsilon }}_{dB}=\frac{1}{N_{p}}\sum_{i=1}^{N_{p}}{\hat{\varepsilon }}_{idB}$$

Absolute radiometric accuracy is defined as the error between the measured and the actual values of the backscattering coefficient for a uniform target, or the radar cross-sectional area for a point target, on the SAR image. The error is usually expressed in dB. The absolute radiometric accuracy measures how well the radiometrically calibrated image pixel reflects the true backscattering coefficient or radar cross-sectional area of the corresponding ground object. Equation (3) gives the formula for calculating the absolute radiometric accuracy:(3)$$\Delta_{rad\_a}=\max\left\{\left|{\hat{\sigma }}_{i}-\sigma_{i}\right|\right\}$$
where ${\hat{\sigma }}_{i}$ is the RCS measurement value (dBm^2^) of the i inspection point and $\sigma_{i}$ is the nominal RCS measurement value (dBm^2^) of the i inspection point.

The system performance in complex environments was tested by applying the radiometric calibration, which only compensated for the effects of the antenna pattern’s cubic term and the distance. The point target energy of the index inspection point (corner reflector), $\varepsilon_{P}$, is calculated using the peak value method. The peak value method takes the amplitude of the peak point in the energy sub-image that corresponds to the inspection point as the value of $\varepsilon_{P}$. The relative and absolute radiometric accuracies of the system were evaluated using multiple calibrators that underwent several voyages. The results showed that the system achieved a relative radiometric accuracy of less than 1 dB and an absolute radiometric accuracy of less than 2.5 dB across the entire swath.
3.Polarization Calibration

Polarization calibration eliminates the distortion of polarization information caused by the polarization coupling and polarization mixing, thus enabling the image to reflect the real polarization scattering characteristics of ground objects. Polarization calibration aims to improve the reliability and consistency of the polarization information of SAR images and to provide effective polarization features for the subsequent polarization decomposition. Polarization isolation refers to the ratio of the energy of the transmitted or received H-polarized (V-polarized) signal to the energy of the serially transmitted or received V-polarized (or H-polarized) signal. The equation for polarization isolation is given by (4):(4)$$\Delta_{pol1}=20\cdot \mathrm{lg}\left(\min\left\{\sqrt{\sum_{i=1}^{N_{p}}\frac{{\left(abs\left(HH_{i}/HV_{i}\right)\right)}^{2}}{N_{p}}},\;\sqrt{\sum_{i=1}^{N_{p}}\frac{{\left(abs\left(VV_{i}/VH_{i}\right)\right)}^{2}}{N_{p}}}\right\}\right)$$

The polarization channel amplitude imbalance is defined as the ratio between the HH and VV signal energies of the co-polarization channels. The amplitude imbalance calculation equation is given by (5):(5)$$\Delta_{pol2}=20\cdot \mathrm{lg}\sqrt{\sum_{i=1}^{N_{p}}\frac{{\left(abs\left(HH_{i}/VV_{i}\right)\right)}^{2}}{N_{p}}}$$

Polarization channel phase imbalance is defined as the difference between the HH and VV signal phases of the co-polarization channels. The phase imbalance calculation equation is given by (6):(6)$$\Delta_{pol3}=\sqrt{\sum_{i=1}^{N_{p}}\frac{{\left(phase\left(HH_{i}/VV_{i}\right)\right)}^{2}}{N_{p}}}$$

We measured the resolution index using multiple calibrators that underwent several passes, and the calibration results are presented in Table 6.

From Table 7, it can be seen that the polarization isolation is better than −30 dB, the amplitude unbalance is better than 1 dB, and the phase unbalance is better than 5° across the entire width range.

## 4. Experiment and Discussion

On 24 and 27 May 2020, agricultural observation experiments were carried out in Yushu, Jilin Province, and S-band fully polarimetric quantitative data were obtained for a total of about 1400 square kilometers (two way). Figure 6 shows the Pauli decomposition image of the same area before and after calibration. Agricultural users achieved 85% accuracy for soil moisture retrieval and straw incorporation. Through further multi-source, multi-temporal, and multi-angle observations, the S-band RCS characteristic curves of different ground objects and various land cover types were counted. We constructed the polarization scattering characteristic matrix and conducted inversion studies of bare soil and vegetated soil parameters, which established a basis for further studying the interaction between S-band electromagnetic waves and land cover types.

### 4.1. Agricultural Application Experiment

According to reference [28], the three coefficients are obtained in (7) by applying Pauli decomposition, which means that the backscatter process in a single station is the coherent superposition of these three mechanisms.
(7)$$\alpha =\frac{S_{\mathrm{hh}}+S_{\mathrm{vv}}}{\sqrt{2}}\beta =\frac{S_{\mathrm{hh}}+S_{\mathrm{vv}}}{\sqrt{2}},\gamma =\sqrt{2}\cdot S_{\mathrm{hv}}$$
where α is a single scattering, β is an even scattering with an incident angle of 0 degrees, γ is an even scattering with a 45-degree inclination angle where all signals are backscattered to the cross-polarized channel and is dominant for volume scattering that produces large cross-polarized echoes. 

We used the S-band SAR system to perform Pauli decomposition of the area (45.1181° N–45.1250° N and 126.5000° E–126.509° E). Figure 7a shows the results, where ${\left|\alpha \right|}^{2}$ is blue, ${\left|\beta \right|}^{2}$ is red, and ${\left|\gamma \right|}^{2}$ is green. Red and blue indicate single and even scattering (horizontal and vertical), blue reflects even scattering on ground and vegetation stems, and green shows volume scattering. Figure 7b shows an optical picture of this area. The red box in Figure 7a contains the black soil and Table 7 shows its backscattering coefficient. The backscattering coefficient obtained in this region is consistent with the mean value given in [29], which indicates that the quantitative accuracy of the S-band radar system can meet the requirements.

Table 7 shows the backscattering coefficients of the full straw-covered land, partial straw-covered land, and sprouting land, among which the sprouting land can be considered as mulch with no straw. Figure 8 displays the measurement point data and the ground synchronous measurements for agricultural users. Moreover, Figure 9a exhibits red dominance, indicating single scattering and odd reflection as the main components with full straw coverage. Blue and green components increase in Figure 9b,c, indicating increased even and odd scattering for dihedral angle scattering when the straw coverage decreases. Therefore, the polarization scattering characteristics enable a qualitative analysis of straw coverage for different ground objects.

In this section, the polarization scattering characteristics and backscattering coefficient measurements of different land types are investigated using the data obtained from the flight field experiment. Figure 9 and Figure 10 show the polarization scattering characteristics and backscattering coefficients of different soils, and Table 7 shows the results. The RCS and backscattering coefficients of the land differ under different coverages. This difference can be used to estimate the straw coverage and provide a basis for the future classification of land types.

Further experiments were carried out to measure the backscattering coefficient of ground objects. Figure 11 shows the backscattering coefficients of bare and straw-covered land.

### 4.2. Multi-Temporal Change Detection and Soil Moisture Inversion

We obtained the polarization scattering characteristic map for the area (45.1451° N–45.1807° N, 126.5154° E–126.5605° E) by performing flight tests on 24 and 27 May 2020. Figure 12a shows the partial area map by Pauli decomposition. Figure 12b shows the agricultural user’s measurement point data. Figure 13 shows the backscattering characteristic curve within the same viewing angle range for this area.

The backscattering coefficient depends on the complex interaction between electromagnetic waves and the ground surface, which involves not only the dielectric constant of soil (mainly related to soil moisture) but also surface roughness, vegetation cover, and other parameters. Hence, the radar backscattering coefficient and soil moisture have an uncertain relationship. Data from multiple observation modes (multi-polarization, multi-angle, and multi-band) can reduce uncertainty and invert soil moisture [2]. This experiment used a quantization system that can perform fully polarimetric, multi-angle, and multi-temporal observations. This system can provide both multi-temporal and multi-angle ground backscattering information for soil water inversion and polarization scattering information for constructing a more accurate soil water inversion model, thus improving the inversion accuracy.

Rain between 5.24 and 5.27 changed soil moisture content, and the S-band backscattering coefficient reflected this change. Figure 13 shows the difference in backscattering coefficients of the same land at different times. We sampled the same location and obtained 1 dB–2 dB differences in backscattering coefficients at different times, indicating soil water content differences. 

The radar backscattering coefficient mainly depends on the soil dielectric constant (related to soil moisture), surface roughness, and vegetation cover. To show the process of how, quantitatively, water quantity changes inversion by the backscattering coefficient, we assume constant surface roughness and vegetation status during two observation periods. Then, we can calculate the soil dielectric constants in these periods using the Alpha approximation model [18]:(8)$$\left(\frac{\sigma_{0}^{\left(2\right)}}{\sigma_{0}^{\left(1\right)}}\right)\approx {\left|\frac{\alpha_{HH/VV}^{\left(2\right)}(\varepsilon_{s},\vartheta )}{\alpha_{HH/VV}^{\left(1\right)}(\varepsilon_{s},\vartheta )}\right|}^{2}$$
(9)$$\begin{array}{l} \left|\alpha_{HH}\left(\varepsilon_{s},\vartheta \right)\right|=\left|\frac{\left(\varepsilon_{s}-1\right)}{{\left(\cos\vartheta +\sqrt{\varepsilon_{s}-\sin^{2}\vartheta }\right)}^{2}}\right|\\ \left|\alpha_{VV}\left(\varepsilon_{s},\vartheta \right)\right|=\left|\frac{\left(\varepsilon_{s}-1)(\sin^{2}\vartheta -\varepsilon_{s}(1+\sin^{2}\vartheta )\right)}{{\left(\varepsilon_{s}\cos\vartheta +\sqrt{\varepsilon_{s}-\sin^{2}\vartheta }\right)}^{2}}\right| \end{array}$$
where $\left(\sigma_{0}^{\left(2\right)}/\sigma_{0}^{\left(1\right)}\right)$ is the backscattering coefficient ratio between two consecutive acquisitions, $\varepsilon_{s}$ is the dielectric constant, and ϑ is the incidence angle. Then, the dielectric mixing model converts the dielectric constant into soil volumetric water content. In this experiment, we measured the backscattering coefficient of the soil surface at different times. We found that it increased by 2 dB after two rainfalls. Based on the calculation process described above, the soil water content increased by about 0.01 cm^3^/cm^3^.

Of course, this is only a simple approximate model. In order to obtain accurate results, it is necessary to consider other measured parameters and simulation analysis, which is the focus research area of multi-angle remote sensing data inversion. This paper focuses on obtaining accurate multi-temporal fully polarimetric SAR data through experiments. This can reduce the soil water inversion uncertainty and provide more information for agricultural monitoring and interpretation.

### 4.3. Discussion

A discussion concerning the presented results is now addressed.

Initially, we discussed the high sensitivity of the SAR signal to the structure and dielectric constant of farmland and crops and the wide use of field physical parameter inversion of short-wavelength multi-polarization SAR.

Then, based on the S-band subsystem of China’s high-resolution aerial system, this paper presents the first agricultural flight application. The research covers two aspects: land detection with different wheat straw coverage and multi-temporal soil change detection. The external calibration flight test detects the function and performance indicators of the system before the flight test. More precisely, radiometric calibration and polarization calibration eliminate the nonlinear distortion and reduce the noise in the image. Hence, they improve the image quality and provide accurate basic data for subsequent data analysis and application. 

Finally, we obtained the RCS and backscattering coefficients of land with different straw coverage in Northeast China using radiometric calibration data. It could qualitatively calculate straw coverage according to the differences. The backscattering coefficients of soil are obtained by combining the multi-temporal data, and the changes in soil moisture can be quantitatively analyzed by inversion. We also obtained the false-color images and backscattering coefficient maps of land with different coverage using the data after polarization calibration. The polarization scattering characteristics of different ground objects differed, providing a basis for land classification. 

Overall, this paper focuses on the design, preparation, and processing of an agricultural flight test. Accurate ground full polarization data were obtained through calibration tests, but a series of factors, such as the depth and area of straw cover and the change in soil roughness, were not measured in detail in the first experiment, so only a relatively simple model was used for approximate calculation. The obtained results were consistent with the trend of the actual measured values, indicating that the S-band could effectively reflect the two kinds of changes after quantitative design, which had a certain enlightening effect on the subsequent agricultural quantitative application.

This experiment demonstrated the potential of an S-band fully polarimetric system for agricultural applications. For further research, the following points should be considered:

1. For the land classification with straw cover, the depth and area of straw cover should be accurately measured. Then, the controlled variable method should be used to conduct multiple comparative experiments to analyze the relationship between the backscattering coefficient and the depth and area of straw cover, combined with the polarization scattering characteristics, to complete the land classification.

2. To calculate the moisture content accurately, the soil roughness change, the optical thickness of soil and vegetation, and various scattering mechanisms should be considered, and a suitable model should be established and combined with optical remote sensing to improve the inversion accuracy.

3. The research on the actual rainfall amount of soil moisture inversion is an important link to verify the theoretical results by practice. A rigorous mathematical model should be established considering factors such as the rainfall intensity and area, soil infiltration rate, soil water holding capacity, soil deep infiltration amount, soil evaporation amount, crop transpiration amount, etc.

## 5. Conclusions

In this paper, we describe the first agricultural application experiment of the S-band sub-system of the high-resolution aerial system in China. We presented the research results of land detection with different wheat straw coverage and multi-temporal soil change. We measured the backscattering coefficient under different straw coverage and water content conditions. The results showed that the backscattering coefficient differed by about 6 dB and 2 dB, respectively (see Figure 11 and Figure 13). We estimated that the soil water content increased by about 0.01 cm^3^/cm^3^. We also analyzed the polarization scattering characteristics of different land types. We found that they were significantly different, which provided a basis for land classification. Overall, this experiment demonstrated the potential of microwave remote sensing for agricultural quantification.

## Figures and Tables

**Figure 1 sensors-24-00236-f001:**
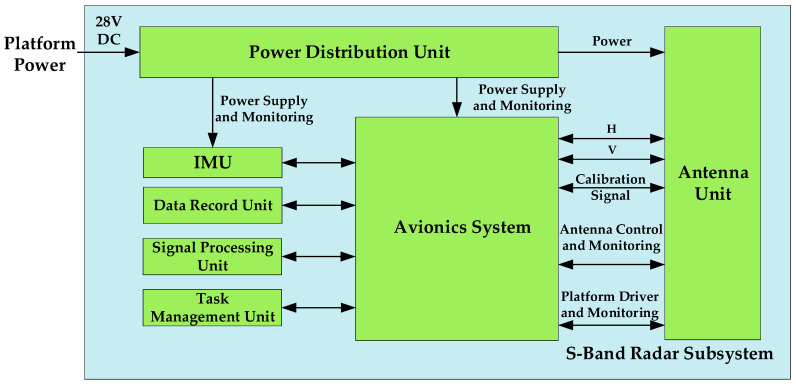
System composition of multi-dimensional joint work.

**Figure 2 sensors-24-00236-f002:**
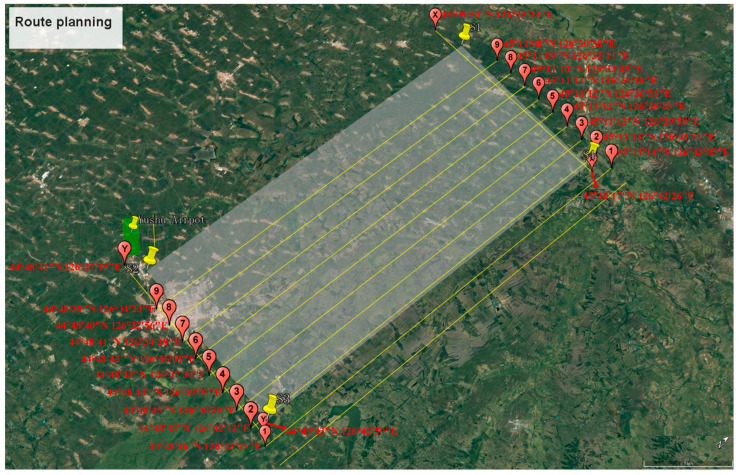
Route planning.

**Figure 3 sensors-24-00236-f003:**
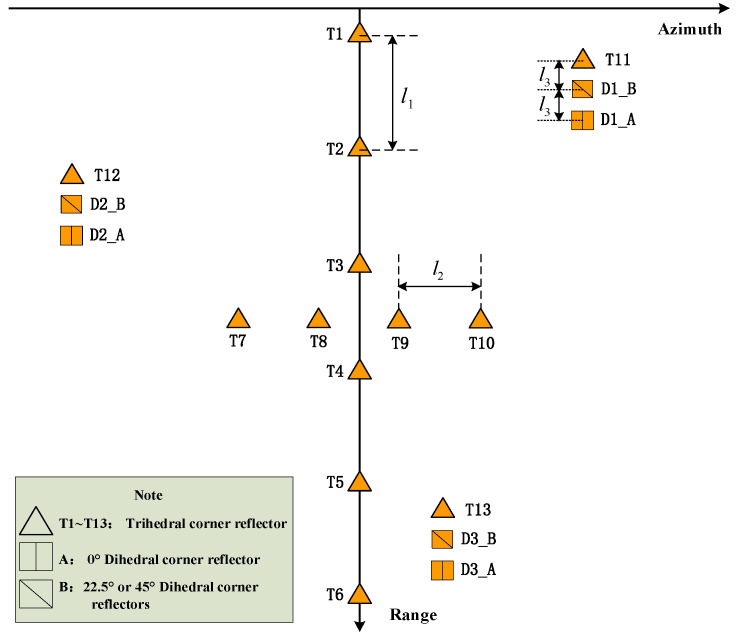
Layout diagram of calibration field.

**Figure 4 sensors-24-00236-f004:**
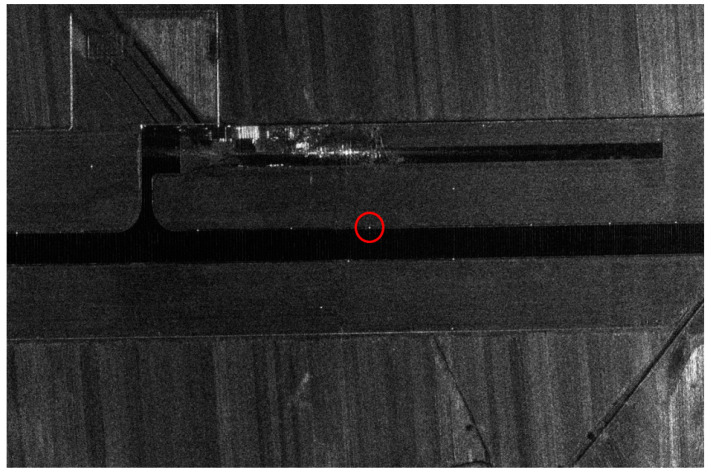
Calibration field imaging result. (The red box is a calibrator).

**Figure 5 sensors-24-00236-f005:**
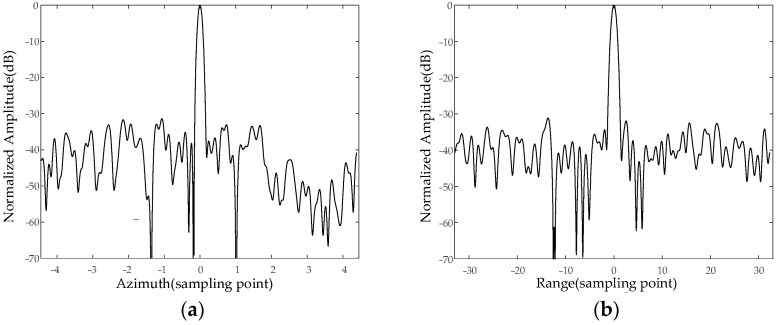
Mode 1 (resolution = 1 m) point target corresponding (midrange) to (**a**) Azimuth and (**b**) range.

**Figure 6 sensors-24-00236-f006:**
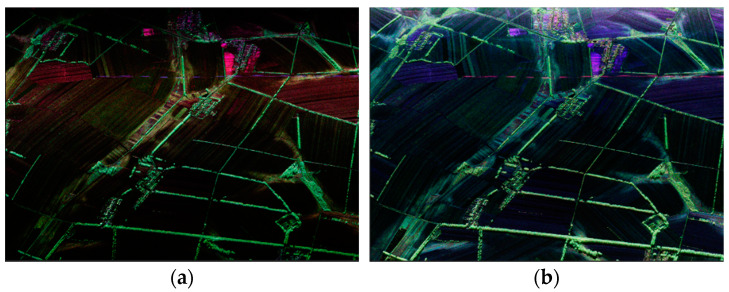
Quantitative measurement effect (Pauli decomposition) (**a**) before the calibration of the elm survey area and (**b**) after the calibration of the elm survey area.

**Figure 7 sensors-24-00236-f007:**
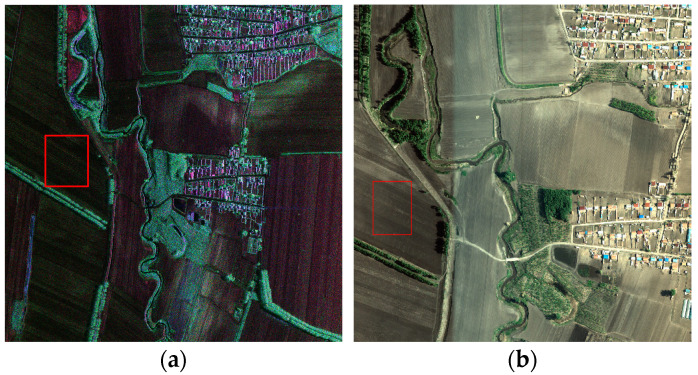
Polarization scattering characteristics of black soil visualized using (**a**) black Earth image (Pauli) and (**b**) optical image.

**Figure 8 sensors-24-00236-f008:**
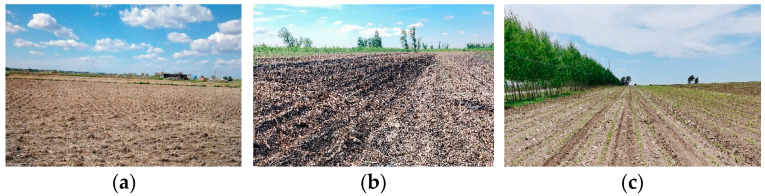
Surveying point maps of the following agricultural users: (**a**) full straw-covered land, (**b**) partial straw-covered land, and (**c**) sprouting land.

**Figure 9 sensors-24-00236-f009:**
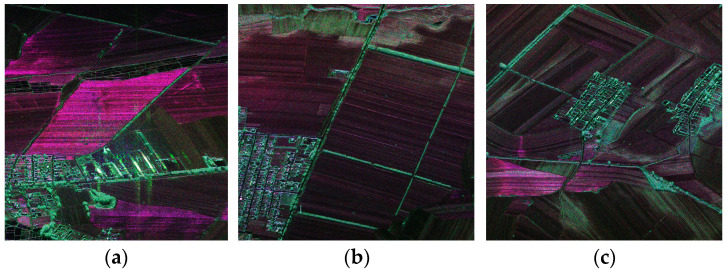
Scattering characteristics of different land polarizations (Pauli decomposition) for (**a**) full straw-covered land, (**b**) partial straw-covered land, and (**c**) sprouting land.

**Figure 10 sensors-24-00236-f010:**
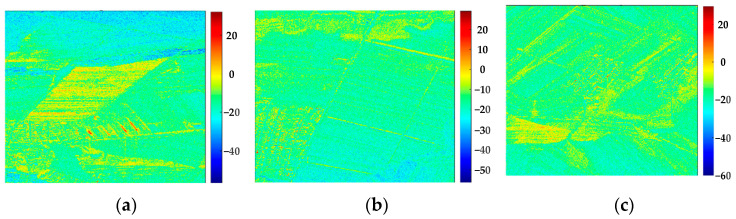
Backscattering coefficient maps of different land types: (**a**) full straw-covered land, (**b**) partial straw-covered land, and (**c**) sprouting land.

**Figure 11 sensors-24-00236-f011:**
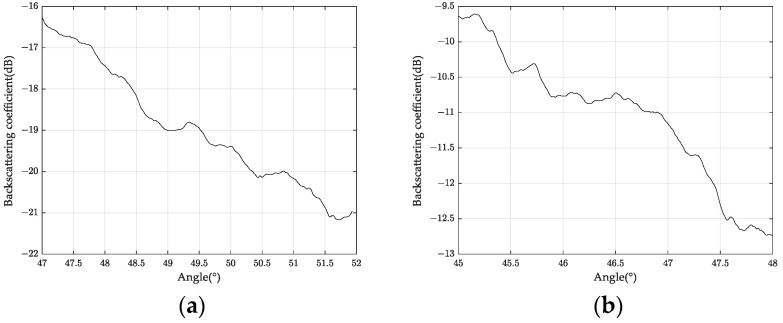
Black soil backscattering coefficients of S-Band for (**a**) bare black Earth and (**b**) straw mulch.

**Figure 12 sensors-24-00236-f012:**
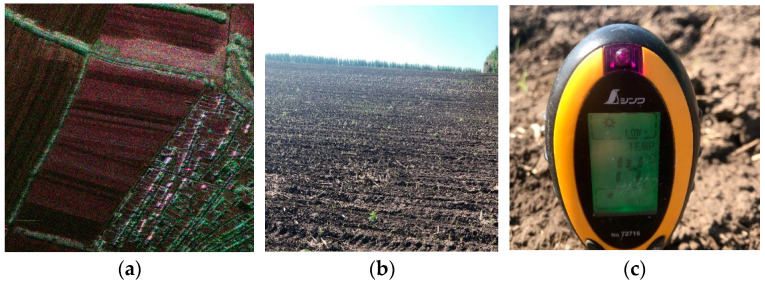
Testing region properties: (**a**) land polarization scattering diagram (Pauli decomposition) and (**b**) user measuring point data (**c**) Local temperature measurement.

**Figure 13 sensors-24-00236-f013:**
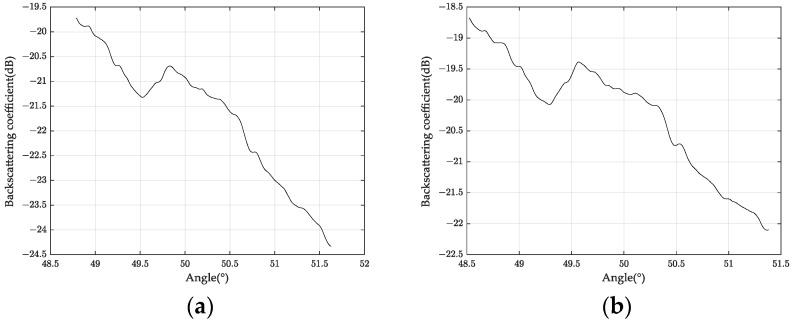
Backscattering coefficient measured in the same area on (**a**) May 24 and (**b**) May 27.

**Table 1 sensors-24-00236-t001:** Technical indexes.

Parameters	Value
Operating frequency	S, Center Frequency 3.2 GHz
Operating pattern	Fully polarimetric stripmap mode
Polarization	HH, HV, VH, VV
Resolution/Swath width/Operating distance	1 m/15 km/20 km
Operating range	1000–10,000 m
Polarization isolation	≥28 dB (Antenna) ≥30 dB (Corner reflector)
Noise equivalent sigma zero	−30 dB (20 km operating distance, 1 m resolution) ≤−33 dB (30 km operating distance, 3 m resolution) ≤−35 dB (40 km operating distance, 10 m resolution)
Side-lobe level and ambiguity	ISLR ≤ −13 dB; PSLR: ≤−20 dB; Azimuth ambiguity ≤ −20 dB; range ambiguity ≤ −20 dB
Quantitative measurement index	
Geometric measurement accuracy	Horizontal accuracy ≤ 5 m (1σ) (Flatland)
Relative radiometric calibration	≤1 dB (3σ)
Absolute radiometric calibration	≤2.5 dB (3σ)
Polarization imbalances	Phase ≤ 5° (1σ); amplitude ≤ 1 dB (1σ)

Note: ISLR: Integral sidelobe ratio; PSLR: peak side lobe ratio.

**Table 2 sensors-24-00236-t002:** Four-point coordinates of the Jilin Yushu survey area.

Vertex	Longitude	Latitude
S1	126.463310	45.171747
S2	126.475753	44.823366
S3	126.707185	44.825671
S4	126.697717	45.173504

**Table 3 sensors-24-00236-t003:** Introduction of the calibration flight.

Index	Calibration	Survey
Task date	2020.0524	2020.0524\0527
Survey area	Jilin, Yushu	
Side visual angle	45° (Left view)	
Flight altitude	2200 m	4200 m
Flight speed	250 km/h	
Swath	1456 m	4000 m
Lateral overlapping degree	\	30%
Lateral extension	\	20%
Mean height of survey area	\	200

**Table 4 sensors-24-00236-t004:** Testing equipment at calibration field.

No.	Description	Qty.	Remarks
1	0° dihedral corner reflector (side length of 0.6 m)	3	Polarization calibration
2	22.5° dihedral corner reflector (side length of 0.6 m)	2	
3	45° dihedral corner reflector (side length of 0.6 m)	1	
7	trihedral angle reflector (right-angle side length of 0.7 m)	13	Resolution measurement and radiation calibration mode
8	Difference measurement equipment	1 set	Measure the position of aerial carrier and the corner reflector

**Table 5 sensors-24-00236-t005:** Radar and flight parameters.

Parameter	Value
Ground elevation (m)	200
Calibration flight elevation (m)	2200
Local elevation (m)	200
Flight speed (m/s)	70
View-angle coverage (°)	27–53
Sampling frequency (MHz)	400
Wave length (m)	0.09375
Average power (w)	19.2
Receiver gain (dB)	67

**Table 6 sensors-24-00236-t006:** Polarization calibration results.

Parameter	Value
Polarization isolation (dB)	−30.67
Phase imbalance (°)	3.41°
Amplitude imbalance (dB)	0.98 dB

**Table 7 sensors-24-00236-t007:** RCS and backscattering coefficients of different land types.

Land Type	RCS (dB)	Backscattering Coefficient (dB)
Full straw mulch earth	−23.1334	−7.4075
Partially covered land with straw	−33.6665	−17.2415
Sprouting land	−32.7417	−15.8347

## Data Availability

Data available on request due to restrictions eg privacy or ethical.

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
