# Peer review of "Agricultural Application Prospect of Fully Polarimetric and Quantification S-Band SAR Subsystem in Chinese High-Resolution Aerial Remote Sensing System"

_sensors, 2023, doi:10.3390/s24010236_

Round 1

Reviewer 1 Report (Previous Reviewer 2)

Comments and Suggestions for Authors

The paper provides a good overview of the use of SAR in agriculture. However, as the reviewer points out, the novelty of the paper is not enough to justify it. Additionally, the authors could improve the paper by adding more detail and explanation to certain sections.

  • Novelty: The authors should add a paragraph to the introduction that discusses the novelty of their work and how it contributes to the existing literature on SAR in agriculture.
  • Moisture measurement: The authors should explain why SAR is a better technology for measuring soil moisture than other methods, such as capacitance, resistance, and inductance sensors. Additionally, they should compare SAR to other remote sensing methods for moisture measurement, such as optical and thermal imagery.
  • Other applications: If the authors determine that SAR can be used for other agricultural applications, they should explain the advantages of SAR over current techniques.
  • Table 4 and Figure 2: The coordinates of Table 4 could be included in Figure 2 to reduce the number of pages and make the information easier for the reader to understand.
  • Figure 5 axis: The axes of Figure 5 should have units.
  • Missing figure number: The authors should correct the missing figure number on line 322.
  • Figures 11 and 13: The authors should correct the reference to Figures 11 and 13 in the conclusion, as these figures do not exist in the paper.
  • The authors could improve the clarity of the paper by using more specific and concise language. For example, instead of saying "SAR can be used to monitor crop growth and development," they could say "SAR can be used to measure crop height, biomass, and leaf area index."
  • The authors could also include more examples and case studies to illustrate how SAR is being used in agriculture today. This would help the reader to better understand the potential of SAR for improving agricultural practices.

Author Response

Reviewer 2 Report (Previous Reviewer 1)

Comments and Suggestions for Authors

I would like to thank the authors for their careful response to my comments in the previous round. Obviously, the theme of the article has been focused and the writing has also been improved. Regarding my concern for innovation, the authors wrote in the response that, "the innovation of this paper is to carry out the first agricultural demonstration application based on the S-band subsystem of Chinese High-resolution Aerial System, and to use the quantitative subsystem to conduct agricultural water content and straw experiments. Therefore, to highlight the first application test, data processing and potential analysis of Chinese high-resolution aviation system, this demonstration application was published on the home page of Chinese Academy of Sciences. This paper focuses on the design of pre-flight calibration test, and the data processing and analysis of the water content and straw experiments." From this perspective, I think this work will be worth publishing in Sensors. However, the authors only expressed how they carried out this work in the abstract, introduction, discussion, and conclusion, and did not clearly indicate the contribution and value of this work in terms of innovation just as they replied in the response. In view of this, I suggest the authors revise the relevant parts of the article to make their innovative contribution clearer. After all, a research paper focuses more on novelty than anything else. In addition, it is still necessary to further improve the writing of the article. There are still some unsmooth expressions, grammar errors and unprofessional terms. For example, the abbreviation SAR is defined several times; "bulk scattering" should be "volume scattering".

Comments on the Quality of English Language

Obviously, the theme of the article has been focused and the writing has also been improved. However, there are still some unsmooth expressions, grammar errors and unprofessional terms. For example, the abbreviation SAR is defined several times; "bulk scattering" should be "volume scattering".

Round 2

Reviewer 1 Report (Previous Reviewer 2)

Comments and Suggestions for Authors

I appreciate the authors' response to my comments. However, there are still some formatting issues that need to be addressed. Please split Table 1 onto a single page, move the caption for Section 3.1 to the same page as the main text, and enhance the clarity of the numbers in Figure 2.

Author Response

Reviewer 2 Report (Previous Reviewer 1)

Comments and Suggestions for Authors

All my comments have been properly treated. No further comment. The manuscript is now ready for publication.

Author Response

Dear reviewer,
Thank you very much for your first two rounds of review comments, which have brought our manuscript up to publication standards.
We wish you all the best in your endeavors.
Yabo Liu and co-authors

This manuscript is a resubmission of an earlier submission. The following is a list of the peer review reports and author responses from that submission.

Round 1

Reviewer 1 Report

Comments and Suggestions for Authors

This paper is dedicated to demonstrating the potential of S-band polarimetric remote sensing in agricultural applications by processing the fully polarimetric SAR (PolSAR) images acquired by the S-band SAR subsystem of the Chinese high-resolution aerial remote sensing system in Yushu City, Jilin Province, China from May 24 to 27, 2020. Overall, the manuscript is easy to understand. Detailed comments are given in the following.

1. The theme of the paper should be focused and consistent. It is expressed by the title, abstract and introduction that the focus of this work should be devoted to the agricultural applications of S-band PolSAR. However, the majority of the manuscript (from Page 4 to Page 11) were arranged to introduce the S-band PolSAR system and the related signal processing, calibration and experimental design. Only several simple Pauli decomposition images and RCS map and curves of some typical targets in the imaging area were briefly provided in the Results Section 3. These limited results are neither quantitative nor fully cover all the agricultural applications listed in abstract, i.e., the vegetation growth, soil moisture content inversion, wheat straw returning and ground feature classification.

2. The innovation of the paper should be improved. The paper neither shows the advance of the S-band PolSAR system in technique and processing, nor demonstrates the advantage and special potential of S-band PolSAR in agricultural applications. The PolSAR agriculture applications provided in the paper have already been achieved in bands such as L, C and P. I cannot see any difference in the S-band results compared to those others.

3. The writing of the paper should be significantly improved. The current manuscript was organized in a very loose structure. Many expressions are not smooth (such as the title!). There are also many grammar errors and unprofessional terms (e.g., ‘full polar’ should be ‘fully polarimetric’, ‘fully polarized SAR’ should be ‘fully polarimetric SAR’, ‘backward scattering’ should be ‘backscattering’, ‘polarization decomposition’ should be ‘polarimetric decomposition’, ‘breadth’ should be ‘swath’….). It seems that the authors even mistakenly wrote their affiliation ‘the Aerospace Information Research Institute’ as ‘‘the Institute of Space and Astronautics’’ in Line 110. I suggest the authors submit the manuscript to professionals in the field for English proofreading.

Comments on the Quality of English Language

The writing of the paper should be significantly improved. The current manuscript was organized in a very loose structure. Many expressions are not smooth (such as the title!). There are also many grammar errors and unprofessional terms (e.g., ‘full polar’ should be ‘fully polarimetric’, ‘fully polarized SAR’ should be ‘fully polarimetric SAR’, ‘backward scattering’ should be ‘backscattering’, ‘polarization decomposition’ should be ‘polarimetric decomposition’, ‘breadth’ should be ‘swath’….). It seems that the authors even mistakenly wrote their affiliation ‘the Aerospace Information Research Institute’ as ‘‘the Institute of Space and Astronautics’’ in Line 110. I suggest the authors submit the manuscript to professionals in the field for English proofreading.

Reviewer 2 Report

Comments and Suggestions for Authors

In this paper, the authors use remote sensing in agriculture. I am so sorry, but the paper needs to improve to be considered for publication in a scientific journal. The paper is monotonous and confusing to read. The introduction material and methods should be shorter, and the results should be longer. The paper's objective needs to be clarified. The conclusions and abstract do not present any numerical result. I encourage the authors to improve the paper. Next, I would like to offer some suggestions to improve the paper.

The abstract is short. Add information about the results and conclusions obtained.

The introduction should be shorter. It should resume in 1 page. In the introduction, you should be able to talk about the problem of the area, the solution proposed, related work, the study's objective, and the paper's structure. The paper’s structure should be presented in the last paragraph of the introduction. To speak about the solution proposed, it is recommended to start the paragraph with the sentence “In this paper.”. In general, the introduction is tedious to read.

The references in the text are incorrectly formatted. The format is [X], where x is the reference number.

Change "Narayanan et al. ..." to "Narayanan et al. [6] ..." the same in the other similar references.

On line 158, avoid using expressions like "following figure." What is the following figure? Figures 2 or 3 are after. Are these the figures referred to in the text? Could you correct this mistake in the rest of the paper?

Table 1 is cut between two pages.

Many references are old (more than ten years). As far as possible, use more current references.

In Table 2, the number of decimals of longitude vertex S1 is less than the other coordinates.

Figures 2 and 3 are similar; these can be merged.

In Table 3, the columns with the same information can be merged to facilitate reading.

Use the same structure to say the coordinates. Tables 3 and 4 have different formats of coordinates units. Table 4 can be merged with Figure 3.

In Figure 6, the axles have no units, and the numbers are small and difficult to read.

The results need more explanation, numerical results, and a critical discussion.

The paper's conclusions need to present results or explain something about the paper.
